# Etiological and epidemiological features of acute respiratory infections in China

Zhong-Jie Li[1,55], Hai-Yang Zhang[2,55], Li-Li Ren[3,55], Qing-Bin Lu [4,55], Xiang Ren[1], Cui-Hong Zhang[1], Yi-Fei Wang[1], Sheng-Hong Lin[1], Xiao-Ai Zhang[2], Jun Li[5], Shi-Wen Zhao[6], Zhi-Gang Yi [7], Xiao Chen[8], Zuo-Sen Yang[9], Lei Meng[10], Xin-Hua Wang[10], Ying-Le Liu[11], Xin Wang[12], Ai-Li Cui[13], Sheng-Jie Lai[1,14,15], Tao Jiang [2], Yang Yuan[2], Lu-Sha Shi[1], Meng-Yang Liu[2], Yu-Liang Zhu[1], An-Ran Zhang[2], Zhi-Jie Zhang[16], Yang Yang [17], Michael P. Ward [18], Lu-Zhao Feng[1], Huai-Qi Jing[12], Liu-Yu Huang[19], Wen-Bo Xu[13], Yu Chen[8], Jian-Guo Wu[11], Zheng-Hong Yuan[20], Meng-Feng Li [5], Yu Wang[21], Li-Ping Wang[1✉], Li-Qun Fang [2✉], Wei Liu [2✉], Simon I. Hay [22,23,56], George F. Gao [21,56], Wei-Zhong Yang[21,56] & The Chinese Centers for Disease Control and Prevention (CDC) Etiology of Respiratory Infection Surveillance Study Team*

Nationwide prospective surveillance of all-age patients with acute respiratory infections was conducted in China between 2009–2019. Here we report the etiological and epidemiological features of the 231,107 eligible patients enrolled in this analysis. Children <5 years old and school-age children have the highest viral positivity rate (46.9%) and bacterial positivity rate (30.9%). Influenza virus, respiratory syncytial virus and human rhinovirus are the three leading viral pathogens with proportions of 28.5%, 16.8% and 16.7%, and *Streptococcus pneumoniae*, *Mycoplasma pneumoniae* and *Klebsiella pneumoniae* are the three leading bacterial pathogens (29.9%, 18.6% and 15.8%). Negative interactions between viruses and positive interactions between viral and bacterial pathogens are common. A Join-Point analysis reveals the age-specific positivity rate and how this varied for individual pathogens. These data indicate that differential priorities for diagnosis, prevention and control should be highlighted in terms of acute respiratory tract infection patients' demography, geographic locations and season of illness in China.

---

[1] Division of Infectious Disease, Key Laboratory of Surveillance and Early-warning on Infectious Disease, Chinese Center for Disease Control and Prevention, Beijing, China. [2] State Key Laboratory of Pathogen and Biosecurity, Beijing Institute of Microbiology and Epidemiology, Beijing, China. [3] Institute of Pathogen Biology, Chinese Academy of Medical Sciences and Peking Union Medical College, Beijing, China. [4] Department of Laboratorial Science and Technology, School of Public Health, Peking University, Beijing, China. [5] Sun Yat-sen University, Guangzhou, China. [6] Yunnan Center for Disease Control and Prevention, Kunming, China. [7] Shanghai Public Health Clinical Center, Shanghai, China. [8] Zhejiang University, Hangzhou, China. [9] Liaoning Provincial Center for Disease Control and Prevention, Shenyang, China. [10] Gansu Provincial Center for Disease Control and Prevention, Lanzhou, China. [11] Wuhan University, Wuhan, China. [12] National Institute for Communicable Disease Control and Prevention, Chinese Center for Disease Control and Prevention, Beijing, China. [13] National Institute for Viral Disease Control and Prevention, Chinese Center for Disease Control and Prevention, Beijing, China. [14] School of Geography and Environmental Science, University of Southampton, Southampton, UK. [15] School of Public Health, Fudan University, Key Laboratory of Public Health Safety, Ministry of Education, Shanghai, China. [16] Department of Epidemiology and Health Statistics, School of Public Health, Fudan University, Shanghai, China. [17] Department of Biostatistics, College of Public Health and Health Professions, and Emerging Pathogens Institute, University of Florida, Gainesville, FL, USA. [18] Sydney School of Veterinary Science, The University of Sydney, Camden, NSW, Australia. [19] The Institute for Disease Prevention and Control of PLA, Beijing, China. [20] Fudan University, Shanghai, China. [21] Chinese Centre for Disease Control and Prevention, Beijing, China. [22] Department of Health Metrics Sciences, School of Medicine, University of Washington, Seattle, WA, USA. [23] Institute for Health Metrics and Evaluation, University of Washington, Seattle, WA, USA. [55]These authors contributed equally: Zhong-Jie Li, Hai-Yang Zhang, Li-Li Ren, Qing-Bin Lu. [56]These authors jointly supervised this work: Simon I. Hay, George F. Gao, Wei-Zhong Yang. *A list of authors and their affiliations appears at the end of the paper. ✉email: wanglp@chinacdc.cn; fang_lq@163.com; liuwei@bmi.ac.cn

A cute respiratory infections (ARIs) are a major worldwide health problem associated with high morbidity and mortality[1]. The World Health Organization (WHO) estimates that ARIs rank as the fourth-highest global cause of mortality, resulting in nearly three million deaths worldwide in 2016 (40 deaths per 100,000)[2]. Among them, acute lower respiratory infections (ALRIs)—including pneumonia and bronchiolitis—have become a leading cause of hospital admissions and in-hospital deaths of young children, especially in low and middle-income countries[3]. The ongoing novel coronavirus (COVID-19) outbreak caused by severe acute respiratory syndrome coronavirus 2 (SARS-CoV-2) that was first reported at the end of 2019, is a public health emergency of international concern, responsible for a growing interest in respiratory tract infections as an issue of public health importance[4]. During the last decade, numerous research studies have reported the detection of respiratory pathogens causing ARIs or pneumonia, with a great diversity in prevalence and pathogen spectrum shown across countries and regions, population demography, years, and seasons[5–8]. Results are made further difficult to compare between studies as diagnostic techniques also vary appreciably.

In China, the surveillance for respiratory infectious diseases has been specifically performed for influenza, measles, mumps, pertussis, tuberculosis, and scarlet fever under the national influenza surveillance network, and China Information System for Diseases Control and Prevention for notifiable infectious diseases[9]. Information was formerly lacking on which pathogens were causal to each of the monitored ARIs. To meet this need, a nationwide project was initiated in 2009 and established a network for the active surveillance of ARIs, which was officially maintained by the Chinese Center for Disease Control and Prevention (China CDC)[10]. Towards the end of 2019, more than a decade of data had been accumulated. In this work, using these unique data, we identify the etiological and epidemiological features of ARI in all ages of the population of China for an extended duration, immediately prior to the COVID-19 pandemic.

## Results

**Study patients**. In total, 233,037 ARI patients were recruited and had samples tested, from which 1930 patients were excluded owing to incomplete data records or for having an inadequate sample volume. Therefore 231,107 cases from 277 sentinel hospitals and 92 reference laboratories in 106 cities, were available for the final analyses (on average 21,010 patients from 147 hospitals/laboratories annually) (Fig. 1A, Supplementary Fig. 1). Their median age was 14 years (interquartile ranges (IQR) 2–44 years, 52.2% were aged <18 years), 53.1% were hospitalized, and 20.2% were diagnosed with pneumonia (Supplementary Table 1). Among them, 110,058 patients were tested for all the eight viral pathogens, 26,757 patients for all the nine bacterial pathogens, and 13,524 patients for all viral and bacterial 17 pathogens (Fig. 1B). Based on these three groups of patients, positivity rates and pathogen spectrum information were compared by age groups and by pneumonia status (hereafter pneumonia vs non-pneumonia).

**Pathogen positive rate**. In total 34.8% (38,265/110,058) of the ARI patients tested for all the eight viral pathogens had at least one positive virus detection, with the highest rate determined in children (46.9%, 19,643/41,921). This dropped to 30.2% (5342/17,663) in 5–17 years old school-age children, 26.9% (8961/33,288) in adults, and 25.1% (4319/17,186) in older people (Table 1). The positive rate was comparable between genders (35.8% in male vs 33.8% in female) and significantly higher in the hospitalized than the outpatients (37.2% vs 30.7%). A total of

25,014 pneumonia patients had all viral pathogens tested, and 9654 (38.6%) had at least one positive detection, higher than that of non-pneumonia (33.6%) (P < 0.001). Significantly higher rates were seen in pediatric patients with pneumonia than non-pneumonia (57.9% vs 42.9%, P < 0.001), with a comparable or lower rate for the other age groups (28.1% vs 30.8% for school-age children, 20.5% vs 28.0% for adults and 22.4% vs 26.4% for older people).

In total, 22.8% (6112/26757) of the ARI patients tested for all the nine bacterial pathogens had at least one positive bacterium detection, with the highest rate determined in school-age children (30.9%, 656/2121), followed by children (23.9%, 2511/10,517), older people (21.3%, 1320/7621), and adults (20.3%, 1625/6498) (Table 1). The positive rate was comparable between gender (22.9% in males vs 22.7% in females, P > 0.05), and significantly higher in the hospitalized than the outpatients (23.7% vs 17.8%). A total of 9395 pneumonia patients had all bacterial pathogen tested, with at least one positive detection in 2851 (30.3%) patients, significantly higher than that of non-pneumonia (18.8%) patients (P < 0.001). Within each age group, higher rates were determined in pneumonia vs non-pneumonia, with the highest difference observed in the school-age children (49.9% vs 19.1%).

Among 13,524 patients with all virus and bacteria tested, the coinfection with over two viruses or bacteria was seen in 23.3% (3157/13,524) of the patients with the highest coinfection rate determined in children (34.3%), followed by school-age children (21.4%), older people (13.1%) and adults (12.3%) (Table 1). Male patients had slightly higher coinfection rates (24.1% vs 22.1%, P < 0.05), as did hospitalized vs outpatients (25.3% vs 11.8%). Among 5900 pneumonia patients with all 17 pathogens tested, the coinfection rate was determined at 28.5%, significantly higher than that of non-pneumonia (19.3%) patients. This pattern was shown in all-age groups.

**Pathogen spectrum**. Based on the proportion of positive detection for viruses, the most frequently determined viral pathogen was influenza virus (IFV) (accounting for 28.5% of total positive detection), followed by respiratory syncytial virus (RSV) (16.8%), human rhinovirus (HRV) (16.7%), human parainfluenza virus (HPIV) (13.1%), human adenovirus (HAdV) (10.3%), human coronavirus (HCoV) (5.8%), human bocavirus (HBoV) (4.6%), and human metapneumovirus (HMPV) (4.1%) (Fig. 2). Further genotyping analysis for IFV showed the predominant role of IFV-A, which accounted for 58.3% of the total genotyped patients, in contrast with IFV-B and IFV-C (29.1% and 0.9%, respectively). Within RSV, RSV-A and RSV-B accounted for 45.5% and 32.9% of the total genotypes. Within genotyped HPIV, the predominant was HPIV-3 (56.4%), followed by HPIV-2 (16.9%), HPIV-1 (16.1%), HPIV-4 (10.5%). Further age-specific analysis revealed RSV was the top pathogen found among children (25.7%), followed by HRV (17.4%), HPIV (15.8%), IFV (14.2%), and HAdV (10.7%). IFV and HRV were the second-ranked viruses among the other three age groups (35% and 17.3% among school-age children, 53.9% and 14.1% for adults, 41.7% and 17.7% for older people), whereas the other ranking of viruses differed among age groups. The genotype ranking of IFV, HPIV, and RSV within age groups largely followed that of total patients, except that RSV-B had a higher proportion in older people.

The top five viruses in pneumonia patients remained the same as non-pneumonia, but, their ranking altered (Supplementary Table 2). For pediatric pneumonia, RSV (28.1%), and HRV (18.3%) remained as the two predominant viruses, whereas the proportion of IFV decreased from 16.5% to 9.77%, dropping behind HPIV and HAdV. For school-age children with pneumonia, HRV had exceeded IFV and HAdV to be the leading

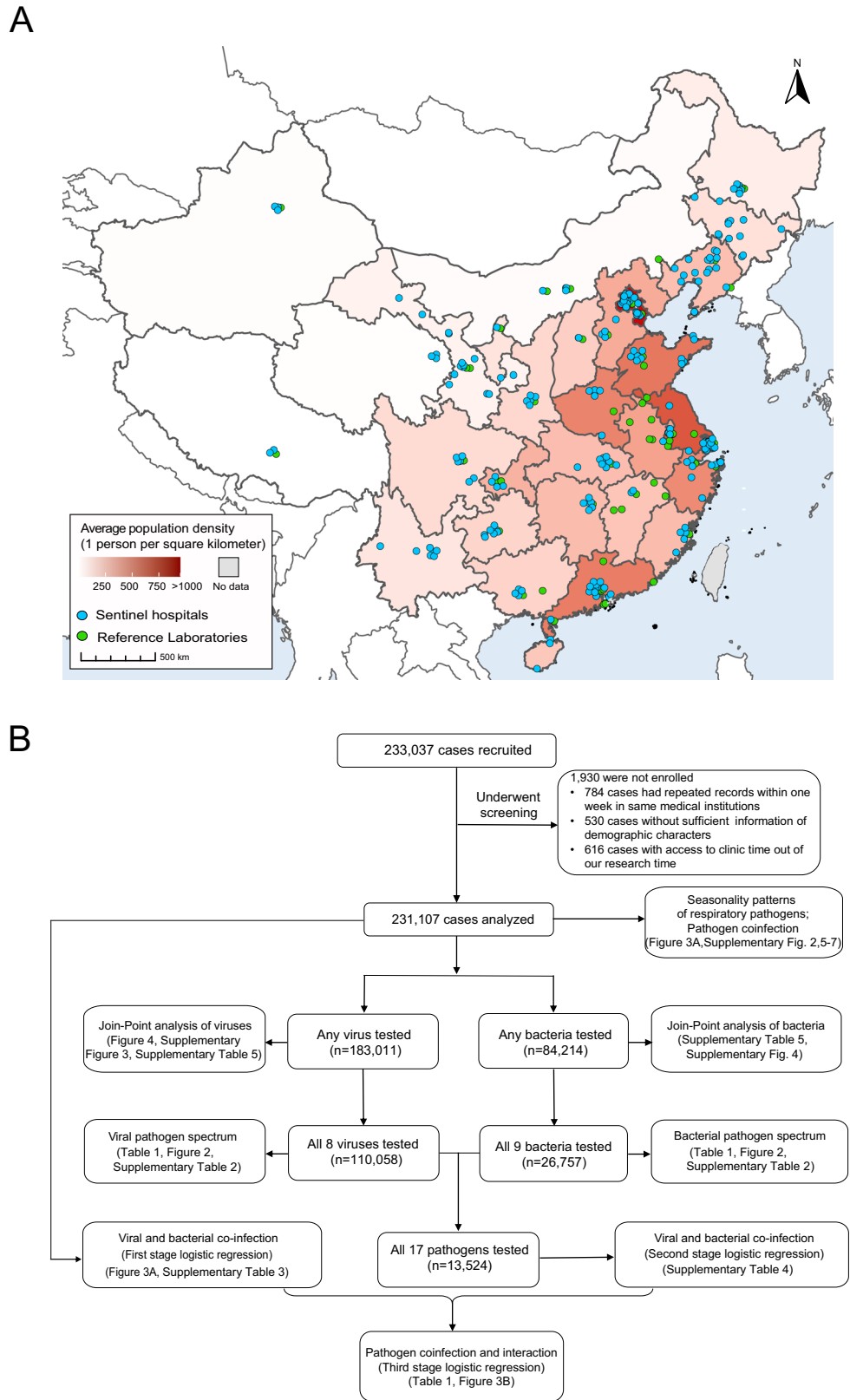

**Fig. 1 Setting of hospital-based surveillance network and data processing in the mainland of China. A** Locations of the 277 sentinel hospitals and the 92 reference laboratories participating in acute respiratory infection (ARI) surveillance from 2009 to 2019. Each point indicates the location of a sentinel hospital (blue) or laboratory (green). The black lines indicated the province boundaries. The background color indicates the average population density of each province from 2009 to 2019 in China. **B** Flowchart of data collection and sorting procedures.

**Table 1 Positive rates of viral and bacterial pathogens and viral–bacterial coinfection rate among patients with an acute respiratory infection (ARI) in the mainland of China, 2009–2019.**

| | All viruses tested | | | All bacteria tested | | | All viruses and bacteria tested[a] | | |
|---|---|---|---|---|---|---|---|---|---|
| | All (N = 110,058) | Pneumonia (N = 25,014) | Non-Pneumonia (N = 85,044) | All (N = 26,757) | Pneumonia (N = 9395) | Non-Pneumonia (N = 17,362) | All (N = 13,524) | Pneumonia (N = 5900) | Non-pneumonia (N = 7624) |
| **All** | 38,265 (34.8) | 9654 (38.6) | 28,611 (33.6) | 6112 (22.8) | 2851 (30.3) | 3261 (18.8) | 3157 (23.3) | 1684 (28.5) | 1473 (19.3) |
| **Sex** | | | | | | | | | |
| Male | 18,980 (35.8) | 5035 (41.3) | 13,945 (34.1) | 3920 (22.9) | 1795 (30.1) | 2125 (19.1) | 2057 (24.1) | 1096 (29.6) | 961 (19.9) |
| Female | 19,285 (33.8) | 4619 (36.0) | 14,666 (33.2) | 2192 (22.7) | 1056 (30.7) | 1136 (18.3) | 1100 (22.1) | 588 (26.7) | 512 (18.4) |
| **Age group** | | | | | | | | | |
| Children (<5 years) | 19,643 (46.9) | 6426 (57.9) | 13,217 (42.9) | 2511 (23.9) | 1193 (30.7) | 1318 (19.9) | 2126 (34.3) | 1220 (37.6) | 906 (30.6) |
| School-age children (5–17 years) | 5342 (30.2) | 1010 (28.1) | 4332 (30.8) | 656 (30.9) | 407 (49.9) | 249 (19.1) | 247 (21.4) | 148 (22.7) | 99 (19.7) |
| Adult (18–59 years) | 8961 (26.9) | 1023 (20.5) | 7938 (28.0) | 1320 (20.3) | 550 (28.4) | 770 (16.9) | 360 (12.3) | 131 (15.8) | 229 (10.9) |
| Older people (≥60 years) | 4319 (25.1) | 1195 (22.4) | 3124 (26.4) | 1625 (21.3) | 701 (25.5) | 924 (19.0) | 424 (13.1) | 185 (15.8) | 239 (11.6) |
| **Case type** | | | | | | | | | |
| Inpatients | 25,513 (37.2) | 8865 (40.2) | 16,648 (35.8) | 5445 (23.7) | 2637 (30.0) | 2808 (19.8) | 2925 (25.3) | 1605 (29.3) | 1320 (21.7) |
| Outpatients | 12752 (30.7) | 789 (26.7) | 11,963 (31.1) | 667 (17.8) | 214 (35.3) | 453 (14.4) | 232 (11.8) | 79 (18.8) | 153 (9.9) |

Number and proportion of at least one positive viral/bacterial pathogen are shown in the table. For viral and bacterial pathogens, the denominators are 110,058 patients with all eight viral pathogens tested and 26,757 patients with all nine bacterial pathogens tested, respectively.
[a]Coinfection number and coinfection rate were calculated based on 13,524 patients with 17 pathogens tested. The coinfection was defined as infected by at least two viral or bacterial pathogens.

pathogen (HRV >IFV>HAdV >HPIV>RSV). For adults and older people, the pathogen spectrum of pneumonia only differed slightly from non-pneumonia, with HPIV up-ranked in both groups (IFV>HRV>HPIV>HAdV>HCoV for adults and IFV>HRV>HPIV>HCoV>RSV for older people). All these patients taken together lead to an altered ranking in pneumonia, with RSV up-ranked from third to first, IFV dropped from first to third, whereas in others the order was unchanged (i.e., RSV>HRV>IFV>HPIV>HAdV) (Fig. 2).

Based on the proportion of positive detection for bacteria, *Streptococcus pneumoniae* (*S. pneumoniae*) was the most frequent bacterium, accounting for 29.9% of the total positive detection, followed by *Mycoplasma pneumoniae* (*M. pneumoniae*, 18.6%), *Haemophilus influenzae* (*H. influenzae*, 15.8%), *Klebsiella pneumoniae* (*K. pneumoniae*, 12.5%), *Pseudomonas aeruginosa* (*P. aeruginosa*, 11.4%), *Staphylococcus aureus* (*S. aureus*, 8.9%), *Chlamydia pneumoniae* (*C. pneumoniae*, 1.6%), *Legionella pneumophila* (*L. pneumophila*, 0.9%) and *Group A Streptococcus* (*GAS*, 0.4%) (Fig. 2). Age-specific heterogeneity was revealed, with the top five bacterial pathogens among children determined as *S. pneumoniae* (38.5%), *H. influenzae* (20.3%), *M. pneumoniae* (18.8%), *S. aureus* (9.7%), and *K. pneumoniae* (7.4%). Among the school-age children, *M. pneumoniae* was up-ranked to the first place, with others slightly altered (*M. pneumoniae*>*S. pneumoniae*>*H. influenzae*>*S. aureus*>*K. pneumoniae*). Among adults and older people, *P. aeruginosa* was up-ranked (i.e., *S. pneumoniae, M. pneumoniae, P. aeruginosa, K. pneumoniae, H. influenzae* in the adults and *S. pneumoniae, P. aeruginosa, K. pneumonia, H. influenzae* and *S. aureus* in older people).

For pediatric pneumonia, the top-listing bacterial pathogens remained the same as that of non-pneumonia, however, with *H. influenzae* up-ranked from third to second (*S. pneumoniae*>*H. influenzae*>*M. pneumoniae*> *S. aureus*> *K. pneumoniae*). For school-age children with pneumonia, there was a remarkable up-ranking of *P. aeruginosa* from eighth to fourth compared with that of non-pneumonia, whereas the other remained unchanged (i.e., *M. pneumoniae*> *S. pneumoniae*>*H. influenzae*> *P. aeruginosa*>and *K. pneumoniae*). For adults with pneumonia, there was an up-ranking of *M. pneumoniae* from sixth to first compared with that of non-pneumonia, whereas the other remained unchanged (*M. pneumoniae* >*S. pneumoniae* > *P. aeruginosa* >*K. pneumoniae* >*H. influenzae*). For older people patients, an identical ranking of the top five bacterial pathogens was observed between pneumonia and non-pneumonia patients (Supplementary Table 2). Notably, when all-age groups were taken together, an identical spectrum and rank of bacterial pathogens were observed between pneumonia and non-pneumonia groups (Fig. 2).

The most frequently seen coinfection was observed among *S. pneumoniae, H. influenzae, M. pneumonia,* and HRV (Fig. 3A), however, the patterns differed across age groups, e.g., the predominant coinfection seen in children was HRV-*S. pneumoniae* and HRV-*H. influenzae*, *S. pneumoniae*-*M. pneumonia* in school-age children, IFV-A-*H. influenzae* in adults and *S. pneumoniae*-*H. influenzae* in older people (Supplementary Fig. 2).

It was notable that most of the interactions referring to IFV-A were negative, predominantly occurring between RSV-A, IFV-A, and RSV-B, between IFV-A and HPIV-3, between IFV-A and HCoV. Positive interaction was identified between HPIV-2 and HAdV, HCoV, and HPIV-3, HBoV and HPIV-3, HMPV, and HCoV. In contrast, most of the viral–bacterial interactions were positive, mainly involving *S. pneumoniae* (with HRV and HCoV) and *K. pneumoniae* (with HRV and HMPV). Among bacteria, *S. pneumoniae* and *M. pneumonia* were less likely to coexist with another bacterium (*S. pneumoniae*-*P. aeruginosa*, *S.*

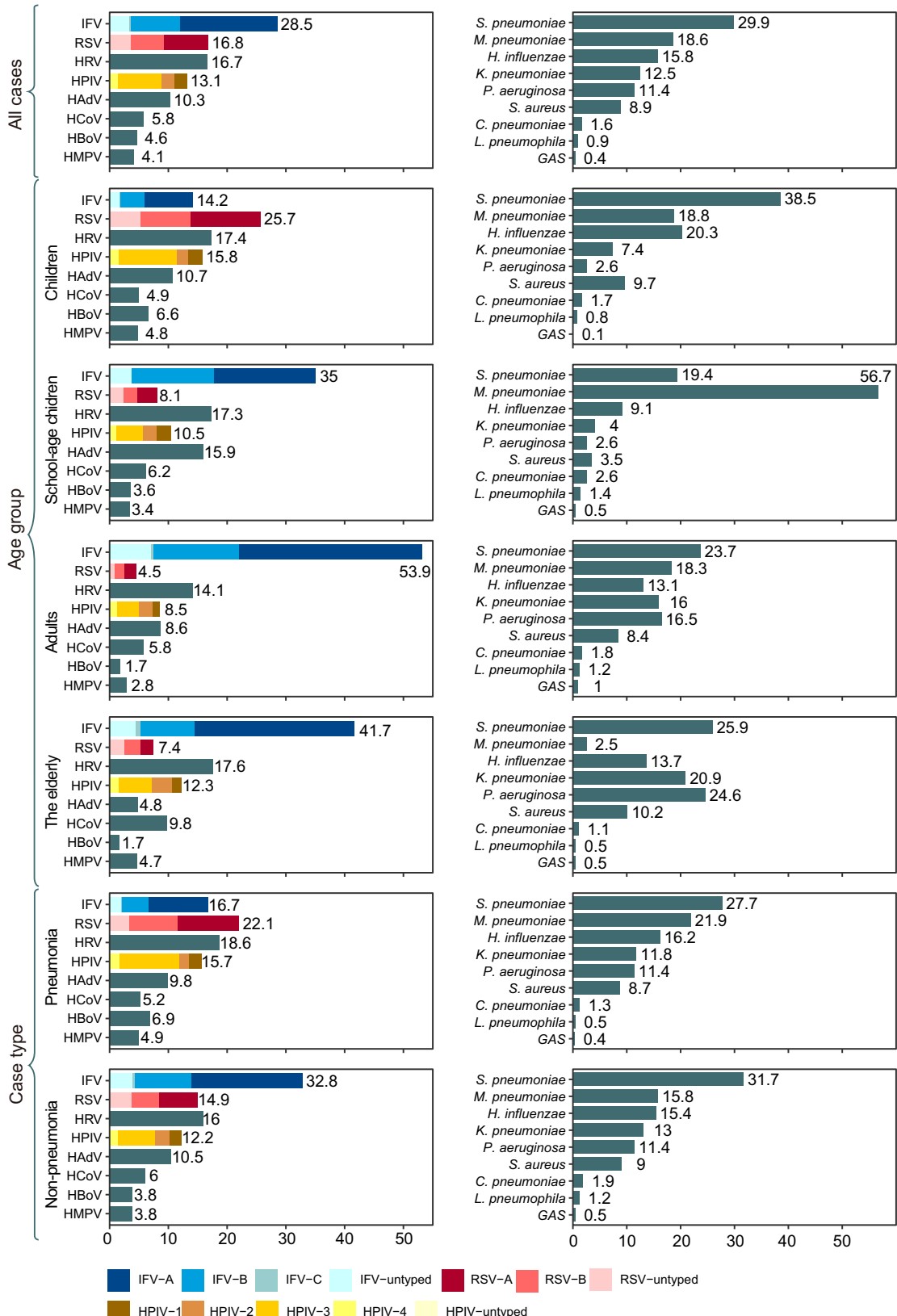

**Fig. 2 Viral and bacterial composition of patients with ARIs in the mainland of China, 2009–2019.** The overall viral composition of 110,058 ARI patients who had all the eight viral pathogens tested. The overall bacterial composition of 26,757 ARI patients who had all the nine bacterial pathogens tested. The length of colored bars and the number behind indicate the proportion of each pathogen, calculated by its positive number used as the numerator and the total positive number of all pathogens used as the denominator. For IFV, RSV, and HPIV, the proportion of their subtypes each colored bar indicated.

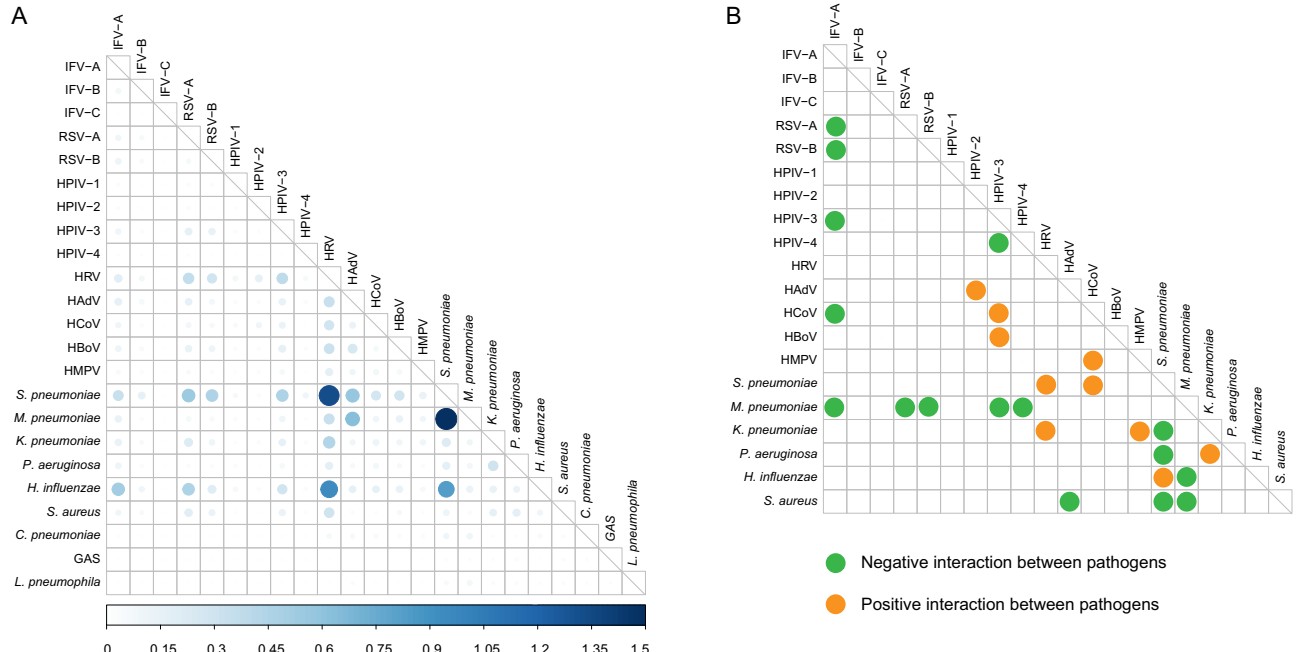

**Fig. 3 Coinfection pattern and interactions of pathogens in patients with acute respiratory infection in the mainland of China, 2009−2019. A** Coinfection rates were calculated pairwise. For pathogen 'X' and 'Y', numerator was the number of patients coinfected both 'X' and 'Y' and the denominator where the total number of patients who were both tested 'X' and 'Y'. Bigger size and darker color of the circles indicate higher coinfection rates between two pathogens. **B** The interactions among pathogens are estimated by host-scale logistic regressions. Positive interactions with two-sided $p$ value <0.05 were denoted in orange and the negative interactions with two-sided $p$ value <0.05 were denoted in green color. The $p$ values were not adjusted for multiple comparisons. The interaction was determined as both significant when without adjusting for multi-pathogens (Supplementary Table 3) and when adjusting for multi-pathogens (Supplementary Table 4) *L. pneumophila, C. pneumoniae,* and *GAS* were not included in the logistic analysis due to the small sample size.

*pneumoniae-K. pneumoniae, S. pneumoniae-S. aureus, M. pneumonia-S. aureus,* and *M. pneumonia-H. influenzae).* The negative viral–bacterial interaction was only observed for *M. pneumonia,* which occurred between *M. pneumonia* and IFV-A, RSV-A, RSV-B, HPIV-3, and HPIV-4) (Fig. 3B, Supplementary Tables 3–4).

**Pattern of age-specific positivity rates.** Altogether, 183,011 patients were tested for at least one of the eight viral pathogens. IFV, as well as its subtype of IFV-A and IFV-B, were observed with higher positive rates in school-age children or adults than in older people or children, while all other viruses were most frequently detected from pediatric cases (Supplementary Table 5). Join-Point regressions (JPRs) revealed the age-specific patterns (Fig. 4 and Supplementary Fig. 3). A child−elderly pattern was shown for RSV (both RSV-A and RSV-B), HRV, and HPIV (mainly in HPIV-1 and HPIV-3), HMPV and HCoV, with the highest rates observed in pediatric patients, followed by one or two descending turn points at ~4 years and 15–18 years (significant APC < 0), with the lowest incidence observed in school-age children or adult group, ensued by increasing turn point (significant APC > 0). The school-age child pattern was featured for IFV, with the highest rates observed in school-age children patients, followed by one descending turn point at about 14 years of age. When subtypes of IFV were separately considered, an earlier turn point at age 6 for IFV-B compared with 14 years old for IFV-A (Supplementary Fig. 3). The child pattern was also featured for HAdV and HBoV, followed by a persistent decrease to the lowest level in older people (Fig. 4).

Altogether, 84,214 patients were tested for at least one of the nine bacterial pathogens. A diversified age-specific pattern was revealed for the presence of tested bacteria (Supplementary

Table 6). A child pattern was revealed for *S. pneumoniae, M. pneumonia, H. influenzae,* and *C. pneumoniae.* The positivity was the highest for children, dropped as age increased, with an obvious descending turn point at nine years for *S. pneumoniae* 7 years for *M. pneumoniae,* and at 6 years for *H. influenza* (Supplementary Fig. 4). Conversely, an elderly pattern was revealed for *K. pneumoniae, P. aeruginosa,* and *S. aureus,* with their positive rate, increased as patients aged. *S. pneumoniae, M. pneumoniae,* and *C. pneumoniae* were more frequently present in inpatients, whereas *P. aeruginosa, K. pneumoniae, S. aureus* and *GAS* were more frequently detected in outpatients (Supplementary Table 5).

**Time trends and seasonality.** During all study years save 2018, IFV remained more frequently detected than other viruses. Three viruses–HRV, RSV, and HPIV remained below IFV, although their annual positive rates varied (Supplementary Fig. 5). Unlike viruses, no consistent trend of positive rates across the years was observed for the tested bacteria. High interannual regularity existed for the variation of circulating subtypes of IFV, RSV, and HPIV. IFV-A and HPIV-3 remained as the major subtype composition, whereas RSV-A and RSV-B alternated as the predominant genotypes.

The seasonal pattern of viruses differed between southern and northern China (Supplementary Fig. 6). Generally, the seasonality was more distinct in northern China. IFV-A circulated in cold months (October to the following January) with one peak in northern China; in contrast, it showed two peaks in January and in August in southern China. The positive rate of IFV-B was similar between northern and southern China, circulating from December to the following March. Even though the epidemic peaks of RSV-A and RSV-B occurred in colder months, the

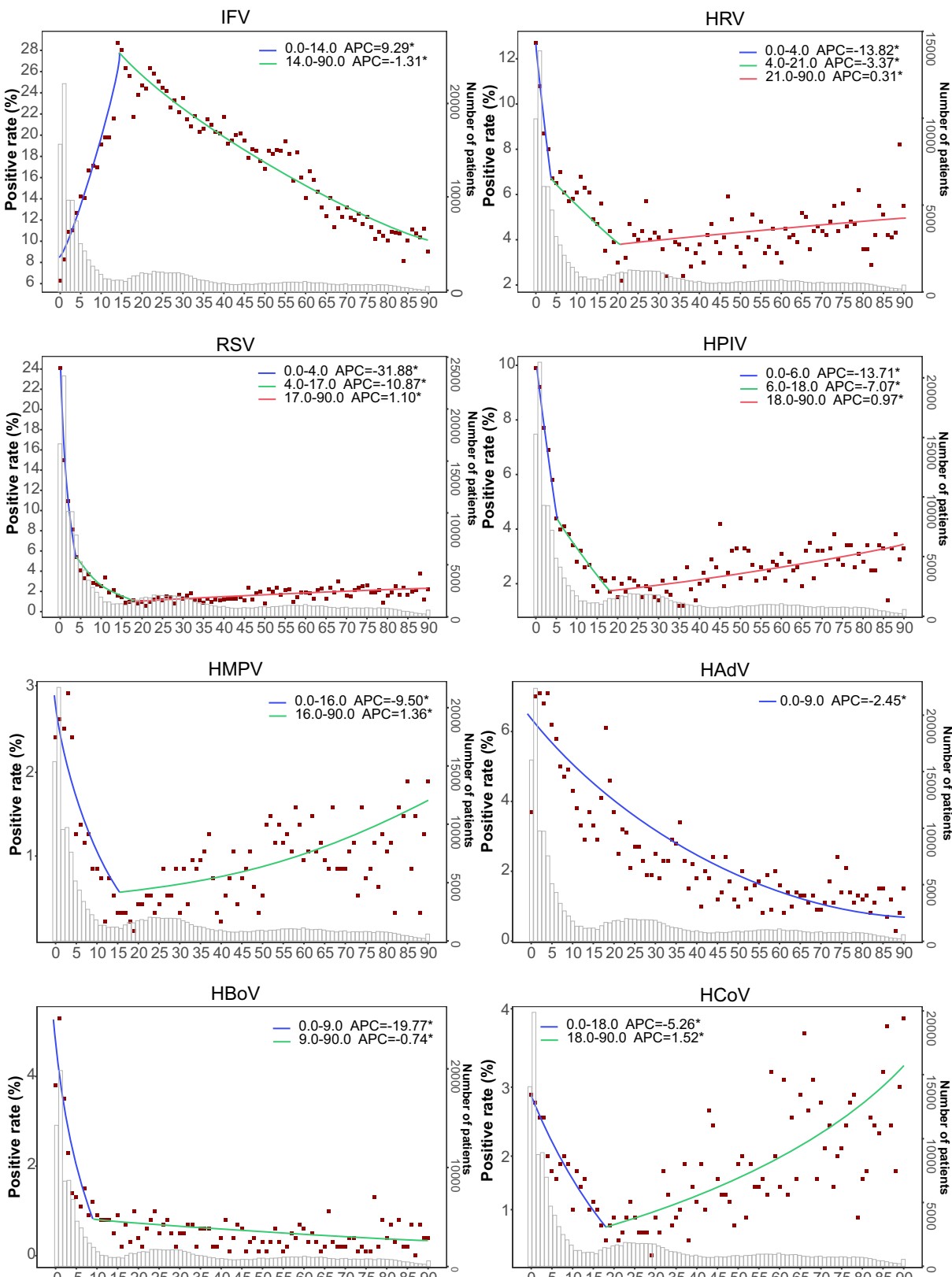

**Fig. 4 The Join-Point regression of the positive rates of each tested virus by age of patient.** A red point indicates the mean positive rate of patients in terms of age and the colored curves indicate fitted patterns by the red points. The colored segments indicate the fitting values of the Join-Point regression. Legends give the Annual Percent Change (APC) value of each fitted curve for each tested virus. *indicates that the APC is significantly different from zero at two-sided $P < 0.05$. The $p$ values were not adjusted for multiple comparisons. The gray bars indicate the number of patients tested for each pathogen.

epidemic period of RSV-B seemed to come earlier and last longer in southern China (Supplementary Fig. 7). The epidemic period of HPIV-1, HPIV-3 and HPIV-4 were the warm months from April to July for both northern and southern China while HPIV-2 were not observed to have epidemic peaks. The HBoV, HCoV, and HRV had higher positive rates in summer but the difference between months was not significant. The HMPV circulated mainly in spring with an epidemic peak in March to April. The HAdV was prevalent throughout the year in northern and southern China.

## Discussion

In this study, by using surveillance data on patients with ARIs that spanned 11 years on all the mainland of China, we have identified viral and bacterial pathogen spectrum and explored their difference in terms of patients' demography.

IFV, RSV, HRV, HPIV, and HAdV were frequently identified in patients with either ARI or pneumonia. Among either ARI or pneumonia patients, RSV was identified as the leading pathogen in the child group, which was consistent with findings from the USA[6]. IFV (mainly IFV-A) and HRV were identified as the leading viral pathogen in both adults and older people, with or without pneumonia. The current study also revealed a higher proportion of HPIV than HMPV in pneumonia in China, as has been shown elsewhere[11].

In addition, by comparing the pathogen spectrum between pneumonia and non-pneumonia patients, we revealed the pathogens that were more likely to be related to pneumonia development if detected in ARI patients of varying age groups. We found an up-ranking of HPIV and *H. influenzae* in pediatric pneumonia[12], of HRV and *P. aeruginosa* in school-age children pneumonia, of HCoV and *M. pneumoniae* among adult pneumonia, and of HCoV in pneumonia of older people. Unexpectedly, HPIV, HRV, and HCoV, which were once thought to cause only a "common cold", were associated with pneumonia. Furthermore, these viruses in combination were responsible for greater annual morbidity than influenza viruses across all-age groups[13,14]. The IFV was shown to be less important in relating to pneumonia, reinforcing the view IFV might act as a significant driver of secondary pneumonia[15].

Our Join-Point analysis results revealed the age threshold that clearly defined the highest-risk group for acquiring infection. Generally, viral pathogens exhibited a consistent age pattern, in that younger than 5 years was at the highest-risk age group, in line with previous studies showing higher detection rates in this age group[6,16]. The exception was IFV, which was detected with the highest rate in school-aged children. Although advanced age is associated with an increase in morbidity and mortality from influenza, school-age children were more frequently infected with influenza, potentially owing to their lower levels of immunity than adults, fewer prior IFV infections, as well as, more opportunities for transmission to occur in highly crowded school settings, when compared with other community settings[17]. Previous studies using data from the national influenza surveillance program concluded similar results from the current study that school-aged children had the highest positive rate of IFV[18–20]. A study from New York City also found that influenza epidemic period peaks occurred earliest among school-aged children each season regardless of circulating influenza viral type, subtype, or strain[21]. In contrast, a more diverse age pattern of infection was discovered for bacteria. Many respiratory bacteria colonize the healthy human respiratory tract, with some causing respiratory infection diseases opportunistically, such as *S. pneumoniae* and *H. influenzae* in children, compared with *K. pneumoniae* and *P. aeruginosa* in older people[22–25]. This was verified by the age turn point determined by

the current Join-Point analysis: 9 years for *S. pneumoniae*, 6 years for *H. influenza*, 40 years for *P. aeruginosa*. However, *K. pneumoniae* was revealed to be more than simply an "elderly" pathogen. Notably, *M. pneumonia* showed the turning point at about 7 years, which is consistent with the highest detection rate in school-aged children documented in England and Wales[26].

Inter-pathogen relationships were hard to distinguish owing to the paucity of simultaneous detection of viral and bacterial pathogens. The viral–bacterial synergistic effects as we found, were mainly from *S. pneumoniae* with respiratory viruses (HRV and HCoV). Competitive interactions of IFV-A with non-influenza virus (RSV-A, RSV-B, and HCoV), as well as between HPIV-3 and HPIV-4 was determined, which are corroborated by previous epidemiological and mechanism studies[27–32]. A range of respiratory pathogens had been suggested to interfere with the growth of the other viruses through resource competition, the immune response, or interference through viral proteins[31,33]. This mechanism can explain the current results showing higher viral–bacterial yet lower viral–viral coinfection rate than expected if a random interactive process was assumed. For example, even for the child group, none of the top five coinfections occurred between viruses; instead between viruses and bacteria. We are clear, however, that the current findings revealed only a statistical relationship between pathogens that perhaps suggest mechanisms or pathways for future research. A better understanding of the complex interactions among viruses, bacteria, and viral–bacterial could help to understand respiratory pathogen epidemiology and planning public health strategies for infection control.

Geographical heterogeneity was identified for the viral activity. In general, southern China was characterized by higher overall positive rates of viruses and relatively lower monthly differences compared with northern China. Epidemic dynamics driven by climate factors (mainly by temperature) partly explained overall detection rates along with seasonality patterns of respiratory viruses and these relationships varied with latitude[34,35].

Our study was subject to several important limitations. First, we failed to identify a detectable etiology in >50% of cases. This limitation seemed to be common to surveillance studies with similar study designs[5,6]. Several factors such as the usage of antibiotics or antiviral drugs prior to treatment might offer plausible explanations for these low detection rates. Second, causal relationships cannot be determined merely from the positive detection of pathogens. Especially for bacterial pathogens, the culture on nasopharyngeal aspirate or sputum is likely to be colonization rather than an invasive infection, which was owing to the clinical dilemma that poorly tolerated invasive tests are usually were not prescribed on patients with mild symptoms or children[6]. Conversely, there is growing evidence showing an association between bacterial colonization levels and future occurrence of ARIs[36].

Despite these concerns, the determination of age-, gender-, temporal-, and spatial-specific pathogen spectra may be helpful in identifying predominant respiratory pathogen candidates for (i) applying differential diagnosis, (ii) prevention control, and (iii) the administration of antivirals or vaccines when available. Unfortunately, vaccines against all of these pathogens are not yet included in planned immunization in China. Commercially available vaccines are provided as an option. Coverage rates of viral or bacterial vaccines were unsurprisingly low, even for the IFV and *H. influenza*, recognized to cause a high disease burden yet showing several immediate opportunities for improved public health intervention[37–39].

## Methods

**Hospital-based surveillance network**. Between January 2009 and December 2019, active surveillance of patients with ARIs was conducted in the 31 provinces

(autonomous regions or municipalities) in the mainland of China, under the management of China CDC. Sentinel hospitals were selected in each province, to sample admissions to general hospitals, children's hospitals, as well as, urban and rural community health service centers (Fig. 1A). The number of hospitals was determined in proportional to the total population size by province. Within each province, sites were selected with high coverage of medical service, adequate capacity for surveillance and laboratory testing. All participating hospitals and laboratories used a standard operating protocol (SOP) of surveillance that included guidelines for patient enrollment, specimen collection, laboratory testing, data recording, and management, developed by China CDC[40]. All the sentinel hospitals had undergone training to be qualified for recruiting patients, sample collection, and test following the SOP, before the surveillance was started. The reference laboratories were responsible for conducting comprehensive pathogenic tests on the specimens. Pre-study training, whole-procedure supervision, monthly enrollment reports, data audits, and annual study-site visits were conducted to ensure uniform procedures were employed as guided among the study sites and across the years. The SOP was approved by the institutional review board at each institution and at the CDC, and remained unchanged across the surveillance years[40,41].

**Patient enrollment and specimen collection.** The ARIs were defined as (1) at least one of the following conditions: fever, abnormal white blood cell (WBC) differentials, leukocytosis or leukopenia; (2) at least one of the following symptoms/signs: cough, chills, expectoration, nasal congestion, sore throat, chest pain, tachypnea, and abnormal pulmonary breath sounds. Pneumonia was diagnosed referring to the Chinese Thoracic Society (CTS) guidelines[42], with minor modification. Those with a chest radiograph demonstrating punctate, patchy, or uniform density opacity were defined as having radiographic evidence of pneumonia. The clinical diagnostic criteria for pneumonia was 1) presence of clinical manifestations of pneumonia: (1) new onset of cough or expectoration, or aggravation of existing symptoms of respiratory tract diseases, with or without purulent sputum, chest pain, dyspnea, or hemoptysis; (2) fever; (3) signs of pulmonary consolidation and/or moist rales; (4) peripheral WBC $> 10 \times 10^9/L$ or $< 4 \times 10^9/L$, with or without a left shift, together with 2) chest radiograph showing new patchy infiltrates, lobar or segmental consolidation, ground-glass opacities, or interstitial changes, with or without pleural effusion. A total of 95,534 cases were hospitalized and 60,104 cases accepted chest radiograph examination. Owing to the samples collected prior to therapeutic measures according to the SOP of ARI surveillance, the patients with pneumonia were not healthcare-associated pneumonia getting from sentinel hospitals in this study. Each sentinel hospital enrolled ARI patients from the department of internal medicine, emergency department, fever department, pneumology department, or infectious diseases department to collect patients according to the preconcerted sample size regulated by the SOP of surveillance[36]. Patients with confirmed diagnoses of non-infectious respiratory diseases such as asthma and respiratory tumor were excluded. Both outpatients and hospitalized patients were recruited according to their admission diagnosis of ARI with pneumonia or not. The respiratory specimens—including nasopharyngeal swab and aspirate, sputum, bronchoalveolar lavage fluid (BALF), pleural effusion, and blood—were collected prior to therapeutic measures, and tested within 24 h of the collection; otherwise, samples were stored at −70°C until tested. If the long-distance transfer was needed, all of the samples were transferred in dry ice, until the arrival at the reference laboratory, and the subsequent detection was performed. Specimens for bacteria cultured should be separated immediately and should not be separated after storage.

National Health Commission of the People's Republic of China decided that since data from patients with ARIs was part of continuing public health surveillance and implemented national surveillance guidelines; parents/guardians of participants in this study were only required to provide brief verbal consent during their enrollment, which was recorded in each questionnaire by their physicians.

**Laboratory procedures.** Nasopharyngeal specimens were collected for detection of eight viral pathogens—IFV, RSV, HPIV, HMPV, HCoV, HRV, HAdV, and HBoV —by reverse transcriptase-polymerase chain reaction (RT-PCR) or PCR[41]. IFV, RSV, HPIV, HMPV, HCoV, and HRV were tested by RT-PCR, and HAdV and HBoV were tested by performing PCR according to the SOP. For IFV, RSV, and HPIV positive samples, the subtypes were further determined by real-time RT-PCR using specific primers and probes (Supplementary Table 6). Automated nucleic acid extraction equipment (i.e., from Roche/Qiagen/bioMerieux/Applied BioSystem company), commercial kits (i.e., Invitrogen/Roche/Qiagen/Promega/Takara), and other traditional methods were all acceptable for nucleic acid extraction. The primer/probes used and amplification conditions are available on request. All steps of the nucleic acid extraction and PCR test were conducted in parallel with positive and negative controls.

Bacterial culture was applied to test blood, nasopharyngeal aspirate, sputum, BALF, and pleural effusion to detect *S. pneumoniae, S. aureus, K. pneumoniae, P. aeruginosa, GAS, H. influenzae*. The extraction nucleic acid and PCR were used on the bacterial culture-negative specimens to detect the above six bacteria and *L. pneumophila, M. pneumoniae,* and *C. pneumoniae*. Urine was also collected for rapid antigen detection for *L. pneumophila* and *S. pneumoniae* detection (details in Supplementary Information).

The primers and sequence information for PCR used in ARIs are listed in Supplementary Figs. 8, 9 and Supplementary Table 6.

**Data management and statistical analysis.** Data on individual demography, clinical manifestations, laboratory testing results, chest radiographic findings, medication use, and outcomes were collected by reviewing medical records and the data were entered into a standardized database by trained clinicians. All the data were uploaded to the online management system structured by the China CDC, sorted to remove redundant data, and checked for incomplete data. Data collection was considered to be public health surveillance by the National Health Commission of the People's Republic of China and verbal informed consent was obtained from patients or their legal guardians. The surveillance protocol was reviewed and approved by the ethics review committees of the China CDC (2015-025).

Four age groups were defined: children (<5 years old), school-age (5–17 years old), adult (18–59 years old), and older people group (≥60 years old). Provinces located in southern and northern China were defined according to latitude (Supplementary Fig. 6). Descriptive statistics included frequency analysis for categorical variables, medians, and IQR for continuous variables. Pearson's Chi-square test or Fisher's exact test were performed to compare categorical variables between groups. The JPR model was used to describe trends that were not constant over ages (V.4.7.0.0, Statistical Research and Applications Branch, National Cancer Institute, USA). Multi-stage logistic regression models were applied to infer pathogen interactions at the individual host level after adjusting for patients' sex, age, illness severity, and disease seasonality (details in the Supplementary Information). All the statistical analysis was performed using R version 3.6.3. A two-sided $p$ value of <0.05 was considered statistically significant. Further details of all laboratory procedures and statistical analyses are provided in the Supplementary Information.

**Reporting summary.** Further information on research design is available in the Nature Research Reporting Summary linked to this article.

## Data availability

Relevant data that support the findings of this study and model results generated as part of this study are publicly available within the paper and its Supplementary Information Files. Raw data are not publicly available and are protected due to data privacy laws, which were used under license for the current study, but are available upon reasonable request to the corresponding author and with permission from the data provider (Li-Ping Wang). The request will be responded within 1 week. Source data are provided with this paper.

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

## Acknowledgements

The authors thank all the subjects, their families, and collaborating clinicians for their participation. We thank staff members of the Department of Science and Education of the National Health Commission, and the ARIs surveillance network laboratories and sentinel hospitals in the participating 31 provinces of China for assistance with field investigation, administration, and data collection. This work was supported by grants from the China Mega-Project on Infectious Disease Prevention (grant number 2018ZX10713001, 2018ZX10713002, 2018ZX10201001, and 2017ZX10103004), and the National Natural Science Funds (grant number 91846302, 81825019). The funders had no role in the design and conduct of the study; collection, management, analysis, and interpretation of the data; preparation, review, or approval of the manuscript; and decision to submit the manuscript for publication.

## Author contributions

W.Z.Y., G.F.G., L.P.W., W.L., and L.Q.F. conceived, designed, and supervised the study. W.Z.Y., Z.J.L., J.L., M.F.L., S.W.Z., Z.G.Y., Z.H.Y., X.C., Y.C., Z.Z., Z.S.Y., L.M., X.H.W., Y.L.L., J.G.W., L.L.R., L.Y.H., H.Q.J., X.R., C.H.Z., Y.F.W., S.H.L. X.A.Z., X.W., S.J.L., T.J., Y.W., A.L.C., and W.B.X. formulated the protocols, guidelines, and SOP of the active sentinel pathogenic surveillance. H.Y.Z., Q.B.L., Y.Y., M.Y.L., A.R.Z., Z.J.Z., S.L.S., Y.L.Z. collected, cleaned, and analyzed the data. L.Q.F., W.L., and L.P.W. wrote the drafts of the manuscript. L.Z.F., Z.J.L., W.Z.Y., S.I.H., G.F.G. interpreted the findings. W.Z.Y., M.F.L., Z.H.Y., Y.C., J.G.W., Y.Y., M.P.W., S.I.H. commented on and revised drafts of the manuscript. All authors read and approved the final report.

## Competing interests

The authors declare no competing interests.

## Additional information

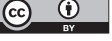

## The Chinese Centers for Disease Control and Prevention (CDC) Etiology of Respiratory Infection Surveillance Study Team

Wei-Zhong Yang[24], George F. Gao[24], Zhong-Jie Li[1], Li-Ping Wang[1], Xiang Ren[1], Yi-Fei Wang[1], Sheng-Hong Lin[1], Cui-Hong Zhang[1], Meng-Jie Geng[1], Xin Wang[12], Huai-Qi Jing[12], Wen-Bo Xu[13], Ai-Li Cui[13], Yu-Juan Shen[25], Yan-Yan Jiang[25], Qiao Sun[26], Li-Peng Hao[26], Chu-Chu Ye[26], Wei Liu[2], Xiao-Ai Zhang[2], Liu-Yu Huang[19], Yong Wang[19], Wen-Yi Zhang[19], Ying-Le Liu[11], Jian-Guo Wu[11], Qi Zhang[11], Wei-Yong Liu[27], Zi-Yong Sun[27], Fa-Xian Zhan[28], Ying Xiong[29], Lei Meng[10], De-Shan Yu[10], Chun-Xiang Wang[30], Sheng-Cang Zhao[30], Wen-Rui Wang[31], Xia Lei[31], Juan-Sheng Li[32], Yu-Hong Wang[33], Yan Zhang[33], Jun-Peng Yang[34], Yan-Bo Wang[34], Fu-Cai Quan[35], Zhi-Jun Xiong[35], Li-Ping Liang[36], Quan-E Chang[36], Yun Wang[37], Ping Wang[37], Zuo-Sen Yang[9], Ling-Ling Mao[9], Jia-Meng Li[38], Li-Kun Lv[38], Jun Xu[39], Chang Shu[39], Xiao Chen[8], Yu Chen[8], Yan-Jun Zhang[40], Lun-Biao Cui[41], Kui-Cheng Zheng[42], Xing-Guo Zhang[43], Xi Zhang[44], Li-Hong Tu[44], Zhi-Gang Yi [7], Wei Wang[7], Shi-Wen Zhao[6], Xiao-Fang Zhou[6], Xiao-Fang Pei[45], Tian-Li Zheng[45], Xiao-Ni Zhong[46], Qin Li[47], Hua Ling[47], Ding-Ming Wang[48], Shi-Jun Li[48], Shu-Sen He[49], Meng-Feng Li [5], Jun Li[5], Xun Zhu[5], Chang-Wen Ke[50], Hong Xiao[50], Biao Di[51], Ying Zhang[51], Hong-Wei Zhou[52], Nan Yu[52], Hong-Jian Li[53], Fang Yang[53], Fu-Xiang Wang[54] & Jun Wang[54]

[24]Chinese Center for Disease Control and Prevention, Beijing, China. [25]National Institute of Parasitic Diseases, Chinese Center for Disease Control and Prevention, Shanghai, China. [26]Center of Disease Prevention and Control in Pudong New Area of Shanghai, Shanghai, China. [27]Tongji Hospital, Tongji Medical College, Huazhong University of Science and Technology, Wuhan, China. [28]Hubei Provincial Center for Disease Control and Prevention, Wuhan, China. [29]Jiangxi Provincial Center for Disease Control and Prevention, Nanchang, China. [30]Qinghai Provincial Center for Disease Control and Prevention, Xining, China. [31]Inner Mongolia Autonomous Region Comprehensive Center for Disease Control and Prevention, Hohhot, China. [32]Lanzhou University, Lanzhou, China. [33]Lanzhou Center for Disease Control and Prevention, Lanzhou, China. [34]Baiyin Center for Disease Control and Prevention, Baiyin, China. [35]Tianshui Center for Disease Control and Prevention, Tianshui, China. [36]Wuwei Center for Disease Prevention and Control, Wuwei, China. [37]Qingyang Center for Disease Control and Prevention, Qingyang, China. [38]Tianjin Center for Disease Control and Prevention, Tianjin, China. [39]Heilongjiang Provincial Center for Disease Control and Prevention, Harbin, China. [40]Zhejiang Center for Disease Control and Prevention, Hangzhou, China. [41]Jiangsu Provincial Center for Disease Control and Prevention, Nanjing, China. [42]Fujian Center for Disease Control and Prevention, Fuzhou, China. [43]Beilun People's Hospital, Ningbo, China. [44]Shanghai Municipal Center for Disease Control and Prevention, Shanghai, China. [45]Sichuan University, Chengdu, China. [46]Chongqing Medical University, Chongqing, China. [47]Chongqing Center for Disease Control and Prevention, Chongqing, China. [48]Guizhou Center for Disease Control and Prevention, Guiyang, China. [49]Sichuan Province Center for Disease Control and Prevention, Chengdu, China. [50]Guangdong Provincial Center for Disease Control and Prevention, Guangzhou, China. [51]Guangzhou Municipal Center for Disease Control and Prevention, Guangzhou, China. [52]Zhujiang Hospital, Southern Medical University, Guangzhou, China. [53]Jinan University, Guangzhou, China. [54]The Third People's Hospital of Shenzhen, Shenzhen, China.

