## [Peer Review File · Nature Communications]

Reviewers' comments:

Reviewer #1 (Remarks to the Author):

Here the authors present the results from a national acute respiratory surveillance program conducted in China from 2009-2018. The results of PCR detection of viral pathogens and bacterial culture and PCR for bacterial pathogens. The authors describe the epidemiology of viral and bacterial infection by age and geographic location and look at associations of coinfection and temporal patterns (i.e. seasonality) of infection in different regions. The manuscript is generally well written and the analyses presented are appropriate.

There are, however, major weaknesses in the manuscript. There is no genetic analysis included, despite sequencing being used as part of the diagnostic pipeline. As such we don't know anything about subtype diversity, e.g. flu A (H1, H3) vs fluB, or HPIV type, etc. As this is a purely epidemiology study it is of some interest but would be better suited to a specialist journal, particularly as large swathes of the paper are descriptive. Another serious limitation is the issue of commensal versus pathogenic bacterial infection, which the authors do not adequately address beyond recognizing it as a limitation in the discussion. But the fact that no causal relationship can be determined between bacterial infection and disease, particularly in the case of bacterial culture, is a major problem that affects interpretation of the results and it simply has not been adequately addressed. It also suggests that the results from the bacterial culture should be deemphasized throughout the manuscript.

Other points.

Importantly, apart from the statistical analysis, the study methods are not well described. The laboratory techniques are not adequately described, with a reference to a China CDC manual with no URL provided and the reviewer cannot find it online, and the ecological regions used are not defined. The age groups are also only made clear in the Suppl info.

General statements that over-emphasise the importance of this study and are incorrect are made. For example, from the start of the discussion, "Numerous previous studies have used an etiological surveillance research design to investigate the pathogen spectrum exhibited in patients, but were limited to specific age group, short study duration or only pneumonia cases". This statement is a serious over-simplification and highlights that the discussion is superficial and not put in the context of previous relevant work. The discussion on competitive interactions also over-states the strength of the data – a statistical relationship – presented here.

General – Some of the abbreviations are not defined in the text.

General – What overlap of data presented here is there with reference 9?

Figures 1 and 5 – As stated in the legend, only data from mainland China is presented, so the map inset showing the islands is not required.

Reviewer #2 (Remarks to the Author):

The paper by Li et al conducted a comprehensive overview of the ARI data collected from the nationwide surveillance data in China. This study provides valuable data on the spatiotemporal pattern of different respiratory pathogens across different age groups. The manuscript is well written. Here are some comments for the authors to improve this interesting work:

1. What are the inclusion criteria for patients being tested and reported? Total testing numbers decreased dramatically since 2016 although lab/hospital numbers increased, some explanations are

needed here. The authors used different ways to measure virus activities, for example, weekly (or monthly?) detection rate for temporal trends and overall (?) for age and gender pattern, "virus detection rate" in Supplementary Table (proportion is more appropriate here), "the proportion of positive detections of each pathogen" in Figure 2, and age-standardized detection rate in Figure 5. All these terms are very confusing. It is better to show clearly what are the numerators and denominators of these different terms.

2. The finding that IFV detection rate peaks at 14 years of age is controversial to previous findings that young children and the elderly are well-known high risk groups. Age curve could be skewed due to the small number of specimens collected in some age groups (youth and older adults) from the sentinel network. If the sampling was randomized, age-specific proportion (age-specific positive no/total positive no) could be a better indicator. Or the authors could use weighting to adjust for under-sampling in some age groups.

3. If the authors have adequate clinical data, it is better to classify bacterial pneumonia into community-acquired pneumonia and healthcare associated pneumonia, because these two types are expected to show different age patterns and pathogen spectra. The authors provide detailed information about data collection and case definition, which is good. I am wondering whether there is any guideline about timing of sample collection (like within 3 days after symptom onset)? If the patients were diagnosed with pneumonia one or two weeks after specimen collection, were they listed as pneumonia cases? How to determine primary and secondary infections if a patient was tested positive for multiple pathogens?

4. L188, the results of join-point analysis add interesting findings to the current knowledge about age patterns. But the terms "child pattern" and "adolescent pattern" are misleading as both refer to the turning points not high risk age groups. Maybe "young children" or "children" are better terms used here to be consistent with "child-senior pattern". The authors shall consider using the WHO definition of adolescents (individuals in the 10-19-year age group). Or you can term this 6-18-year group as "school-age children" instead.

5. Figure 5 could be replaced by a heat map to facilitate comparisons between regions. As far as I know, the sentinel network is designed by dividing into northern and southern China, any particular reasons to compare four "ecological regions" (not geographical regions) in this study?

6. Some discussions about vaccination are needed. What are the uptake rates of Haemophilus influenzae type b (Hib) vaccine, flu vaccine and pneumococcal vaccine in China?

Minor comments

1. It is more common to use "older adults" or "the elderly" than "seniors" in research papers.

2. Use the terms consistently, for example, "HCoV" is sometime written as "HCOV".

3. L195, do you mean such gender difference was only observed in children?

4. L197, "all viruses except for IFV" should be "all viruses except IFV"

5. L252, Celsius is missing. Please clarify whether the model considered five meteorological factors (only four listed in Appendix). Did the authors include absolute humidity in the model? Many previous studies have shown AH drives influenza seasonality. In main text, the authors shall mention explicitly only data of four cities are included and some brief information about these four cities will be useful (geographical locations on Figure 1a, climate zone, etc).

Reviewer #3 (Remarks to the Author):

The manuscript "Etiological and epidemiological features of acute respiratory infections in China" by Li and colleagues presents the results of an almost decade long study. The authors present a wealth of interesting data that will be of broad interest, even with the study limitations that are self-identified. Specific comments follow:

- 1) A major limitation in the study is the large percentage of samples for which no etiologic agent was identified. Were there specific parameters that differentiated the identified versus unidentified samples. For example, was there a longer period between symptom onset and testing for the samples for which no agent was identified?
- 2) The data for influenza virus should be split by at least influenza A versus influenza B (these must surely have been differentially identified by the PCR). The nature of these two virus species is known to be different with differential impact on different ages as an example.
- 3) Do the authors have any suggestion as the underlying cause of the rather unusual nature of the IFV crude rate graphs; specifically the increasing rate from 0 to 18 months which is in contract to every other viral pathogen shown.
- 4) In figure 4b, what does the size of the circle denote?

Responses to the reviewers:

Reviewer #1:

1. There are, however, major weaknesses in the manuscript. There is no genetic analysis included, despite sequencing being used as part of the diagnostic pipeline. As such we don't know anything about subtype diversity, e.g. flu A (H1, H3) vs fluB, or HPIV type, etc. As this is a purely epidemiology study it is of some interest but would be better suited to a specialist journal, particularly as large swathes of the paper are descriptive.

[Response] Many thanks for the reviewer's comments. We have supplemented the information of genotypes of IFV (IFV-A, IFV-B, and IFV-C), RSV (RSV-A and RSV-B), and HPIV (HPIV-1, HPIV-2, HPIV-3, and HPIV-4) and comprehensively describe the genotypes of these viruses in the main text (Figs 2–3). The surveillance data and the analyses were also updated to 2019. In addition, two models including the Join-Point regression (JPR) model and multi-stage logistic regression model were applied to examine the age trends of the positive rates of acute respiratory pathogens and to infer pathogen interactions at the individual host level after adjusting for patients' sex, age, illness severity and disease seasonality, respectively. Although largely descriptive this work provides one of the largest studies yet of the diversity of pre-COVID1-9 ARI infection in China.

2. Another serious limitation is the issue of commensal versus pathogenic bacterial infection, which the authors do not adequately address beyond recognizing it as a limitation in the discussion. But the fact that no causal relationship can be determined between bacterial infection and disease, particularly in the case of bacterial culture, is a major problem that affects interpretation of the results and it simply has not been adequately addressed. It also suggests that the results from the bacterial culture should be deemphasized throughout the manuscript.

[Response] We appreciate the reviewer's suggestions, and have focused on the results of viral infections and deemphasized those of bacterial infections throughout the manuscript as advised, in the revised manuscript, which could be seen in the Abstract and Results sections. We also mentioned developed on the issue of commensal versus pathogenic bacterial infection as one of limitations in the Discussion section as: "Second, causal relationships cannot be determined merely from positive detection of the pathogens. Especially for bacterial pathogens, the culture on nasopharyngeal aspirate or sputum is likely to be colonization rather than an invasive infection, which was due to the clinical dilemma that poorly tolerated invasive tests are usually were not prescribed on patients with mild symptoms or children.⁶ On the other hand, there is evidence showing association between bacterial colonization levels and future occurrence of acute respiratory infections.³⁶" (Page 12, Lines 283-289) and that sputum or nasopharyngeal aspirates from patients are still used to test bacteria in ARI-related studies (Jain S et.al., N Engl J Med. 2015. doi:10.1056/NEJMoa1405870).

3. Importantly, apart from the statistical analysis, the study methods are not well described. The laboratory techniques are not adequately described, with a reference to a China CDC manual with

no URL provided and the reviewer cannot find it online, and the ecological regions used are not defined.

[Response] We appreciate the reviewer's valuable suggestion, and have provided the detailed information on the laboratory techniques including the test method for each pathogen in the Supplementary Information (see Supplementary Figure 8, 9 and Supplementary Table 6.). The Join-Point regression (JPR) model and multi-stage logistic regression model used in this study were also introduced in detail in the Supplementary Information (see supplementary method, Data Collection and Statistical Analysis section). In addition, we have changed the regions only with North and South China instead of ecological regions for comparison of the pathogen spectrum, and have provided the definition of both in the main text (Page 14, Line 362-363).

4. The age groups are also only made clear in the Suppl info.

[Response] Thanks for the reviewer's suggestion. We have defined the age groups in the revised Method section (Page 14, Lines 361-362).

5. General statements that over-emphasise the importance of this study and are incorrect are made. For example, from the start of the discussion, "Numerous previous studies have used an etiological surveillance research design to investigate the pathogen spectrum exhibited in patients, but were limited to specific age group, short study duration or only pneumonia cases". This statement is a serious over-simplification and highlights that the discussion is superficial and not put in the context of previous relevant work.

[Response] We appreciated the reviewer's comments, and have removed them from the revised manuscript.

6. The discussion on competitive interactions also over-states the strength of the data – a statistical relationship – presented here.

[Response] Many thanks for the reviewer's correction, and we have revised this sentence accordingly as: "It should be noted, however, the current findings only revealed statistical relationship between pathogens that perhaps suggest mechanism / pathways for future research. The current findings of complex interactions among viruses, bacteria and viral-bacterial could help to understand the virus epidemiology and planning public health strategies for infection control, which also provides many candidates for future research into infection mechanisms." (Page 11, Lines 269-274).

7. General – Some of the abbreviations are not defined in the text.

[Response] Corrected throughout suggested.

8. General – What overlap of data presented here is there with reference 9?

[Response] Thanks for the reviewer's comment. Data on inpatients from 2009 to 2013 (28369 cases accounting for 12.3% of our data used in this study) were presented in the reference 10 "Feng L *et al.*, PLOS ONE, 2015", which has been included in this study.

9. Figures 1 and 5 – As stated in the legend, only data from mainland China is presented, so the map inset showing the islands is not required.

[Response] Corrected as suggested.

Reviewer #2 (Remarks to the Author):

1. What are the inclusion criteria for patients being tested and reported?

[Response] We appreciate this reviewer's comment, and have provided a detailed introduction on the inclusion criteria for patients being tested and reported in the revised manuscript as: "The ARIs were defined as (1) at least one of the following conditions: fever, abnormal white blood cell (WBC) differentials, leukocytosis or leukopenia; (2) at least one of the following symptoms/signs: cough, chills, expectoration, nasal congestion, sore throat, chest pain, tachypnea, and abnormal pulmonary breath sounds. Pneumonia was diagnosed referring to the Chinese Thoracic Society (CTS) guidelines,³⁸ with minor modification (Supplementary method). Due to the samples collected prior to therapeutic measures according to the standard operating protocol (SOP) of ARI surveillance, the patients with pneumonia were not healthcare associated pneumonia getting from sentinel hospitals in this study. Each sentinel hospital enrolled ARI patients from department of internal medicine, emergency department, fever department, pneumology department, or infectious diseases department to collect patients according to the preconcerted sample size regulated by the standard operating protocol (SOP) of surveillance.³⁶ Patients with confirmed diagnosis of non-infectious respiratory diseases such as asthma and respiratory tumor were excluded. Both outpatients and hospitalized patients were recruited. Both outpatients and hospitalized patients were recruited according to their admission diagnosis of ARI with pneumonia or not." (Page 13, Lines 317–331)

2. Total testing numbers decreased dramatically since 2016 although lab/hospital numbers increased, some explanations are needed here.

[Response] Many thanks for the reviewer's comment. The total testing numbers decreased dramatically since 2016–2017 because the two years were the interval between the second and the third five-year implementation cycles of the active surveillance project.

3. The authors used different ways to measure virus activities, for example, weekly (or monthly?) detection rate for temporal trends and overall (?) for age and gender pattern. "virus detection rate" in Supplementary Table (proportion is more appropriate here), "the proportion of positive detections of each pathogen" in Figure 2, and age-standardized detection rate in Figure 5. All these terms are very confusing. It is better to show clearly what are the numerators and denominators of these different terms.

[Response] We appreciate the reviewer's suggestion, and have corrected "detection rate" as "positive rate" throughout the manuscript, which meant that the number of positive patients divided by the number of ARI patients tested (see Supplementary Table 5). "Proportion" in our

manuscript was used to describe the viral/bacterial spectrum, which meant that for each pathogen the numerator was its number of positive cases and denominator was the total number of positive cases for all pathogens. We have provided the numerators and denominators to improve readability as suggested in the revised manuscript (see Page 17, Table1).

4. The finding that IFV detection rate peaks at 14 years of age is controversial to previous findings that young children and the elderly are well-known high risk groups. Age curve could be skewed due to the small number of specimens collected in some age groups (youth and older adults) from the sentinel network. If the sampling was randomized, age-specific proportion (age-specific positive no/total positive no) could be a better indicator. Or the authors could use weighting to adjust for under-sampling in some age groups.

[Response] We appreciate the reviewer's valuable comments, and have checked the data carefully according to the reviewer's suggestion. We found that the number of specimens collected from the sentinel network was enough for the Join-Point analysis with at least 221 cases tested for IFV in each one-year-old group, and the frequency of each one-year-old group had been provided to the revised Figs 4, S3, and S4. By updating the analyses based on the data of subtypes of viruses, we further found that the IFV positive rate peaked at 14 years of age was mostly depended on the IFV-A, while the IFV-B positive rate peaked at 6 years of age (see Supplementary Fig. 3). We also try to present the age-specific proportion (age-specific positive number/total positive number), but which were shown sampling bias due to the non-randomized sampling and more proportion of sampling in children <5 years old according to our surveillance protocol (Figs 4 and S3). In addition, the positive rate of each age along with its standard error were both imported to the Join-Point analysis as two parameters adjusting for bias caused by under-sampling.

Previous studies using data from the national influenza surveillance program (Yu H, et al., PLoS medicine 2013) concluded similar results from the current study that school-aged children had highest positive rate of IFV (Wang Q, Zhang ML, Qin Y, et al. Zhonghua Liu Xing Bing Xue Za Zhi. 2020;41(11):1813-1817. doi:10.3760/cma.j.cn112338-20200318-00375; Zhou L, Yang H, Kuang Y, et al., BMC Infect Dis. 2019;19(1):89. Published 2019 Jan 25. doi:10.1186/s12879-019-3689-9; Fang LQ, Wang LP, de Vlas SJ, et al. Am J Epidemiol. 2012;175(9):890-897. doi:10.1093/aje/kwr411). A study from New York City also found that influenza epidemic period peaks occurred earliest among school-aged children each season regardless of circulating influenza viral type, subtype or strain (Olson, D. R. et al. PLoS medicine, doi:10.1371/journal.pmed.0040247 (2007).). We have cited these references in the revised Discussion section. (Page 11, Lines 246-248)

5. If the authors have adequate clinical data, it is better to classify bacterial pneumonia into community-acquired pneumonia and healthcare associated pneumonia, because these two types are expected to show different age patterns and pathogen spectra.

[Response] Many thanks for the reviewer's suggestions. Due to the samples collected prior to therapeutic measures according to the standard operating protocol (SOP) of ARI surveillance, the patients with pneumonia were not healthcare associated pneumonia getting from sentinel hospitals in this study. We have mentioned this information in the revised manuscript (Page 13, Lines

322-324). It is a good idea for our further studies to different age patterns and pathogen spectra between community-acquired pneumonia and healthcare associated pneumonia in the future.

6. The authors provide detailed information about data collection and case definition, which is good. I am wondering whether there is any guideline about timing of sample collection (like within 3 days after symptom onset)? If the patients were diagnosed with pneumonia one or two weeks after specimen collection, were they listed as pneumonia cases? How to determine primary and secondary infections if a patient was tested positive for multiple pathogens?

[Response] We appreciate the reviewer's comments. Patients were enrolled according to their admission diagnosis of ARI with pneumonia or not based on the standard operating protocol (SOP) of ARI surveillance. The samples were required to be collected prior to therapeutic measures, and tested within 24 hours of collection; otherwise, samples were stored at -70°C until tested. If the patients were diagnosed with pneumonia one or two weeks after specimen collection, they would not be listed as pneumonia cases. In addition, the current observational study design cannot determine primary and secondary infections if a patient was tested positive for multiple pathogens.

7. L188, the results of join-point analysis add interesting findings to the current knowledge about age patterns. But the terms "child pattern" and "adolescent pattern" are misleading as both refer to the turning points not high risk age groups. Maybe "young children" or "children" are better terms used here to be consistent with "child-senior pattern". The authors shall consider using the WHO definition of adolescents (individuals in the 10-19-year age group). Or you can term this 6-18-year group as "school-age children" instead.

[Response] We appreciate the reviewer's correction. The RSV (both RSV-A and RSV-B), HRV, and HPIV (mainly HPIV-1 and HPIV-3), HMPV and HCoV, with highest rates observed in pediatric patients and an increasing trend in the elderly (significant APC > 0), were defined as "*child-elderly pattern*" to indicate that both children and the elderly were more susceptible to infection. We also revised age group "adolescents" into "school-age children" according to the reviewer's suggestion.

8. Figure 5 could be replaced by a heat map to facilitate comparisons between regions. As far as I know, the sentinel network is designed by dividing into northern and southern China, any particular reasons to compare four "ecological regions" (not geographical regions) in this study?

[Response] Many thanks for the reviewer's suggestion, and we have revised our manuscript by comparing the virus activity between northern and southern China instead of the ecological regions.

9. Some discussions about vaccination are needed. What are the uptake rates of Haemophilus influenzae type b (Hib) vaccine, flu vaccine and pneumococcal vaccine in China?

[Response] Many thanks for the reviewer's comments. We added the discussions accordingly as: "Unfortunately, vaccines against all these pathogens were not yet included in planned immunization in China. There are only commercially available vaccines provided as an option.

Coverage rates of viral or bacterial vaccines were unsurprisingly low even for the IFV and *H. influenza* with high disease burden based on several recent studies suggesting several immediate opportunities for improved public health intervention.” (Page 12, Lines 294-297)

Minor comments

1. It is more common to use “older adults” or “the elderly” than “seniors” in research papers.

[Response] We appreciate the reviewer’s correction, and have revised our manuscript accordingly.

2. Use the terms consistently, for example, “HCoV” is sometime written as “HCOV”.

[Response] Corrected as suggested.

3. L195, do you mean such gender difference was only observed in children?

[Response] We removed the comparison between genders in revised manuscript due to the small difference of them.

4. L197, “all viruses except for IFV” should be “all viruses except IFV”

[Response] Corrected as suggested.

5. L252, Celsius is missing. Please clarify whether the model considered five meteorological factors (only four listed in Appendix). Did the authors include absolute humidity in the model? Many previous studies have shown AH drives influenza seasonality. In main text, the authors shall mention explicitly only data of four cities are included and some brief information about these four cities will be useful (geographical locations on Figure 1a, climate zone, etc).

[Response] We appreciate the reviewer’s suggestion, and have removed this section from our manuscript due to the space and content limitations. This suggestion is very helpful for our further studies.

Reviewer #3 (Remarks to the Author):

1. A major limitation in the study is the large percentage of samples for which no etiologic agent was identified. Were there specific parameters that differentiated the identified versus unidentified samples. For example, was there a longer period between symptom onset and testing for the samples for which no agent was identified?

[Response] We appreciate the reviewer’s comments. As we disclosed in revised manuscript, age and pneumonia were related to different positive rates of pathogens. We added comments in Discussion section as: “This limitation seemed to be common to surveillance studies with similar study design. Several factors such as usage of antibiotics or antiviral drugs prior to treatment offer plausible explanation for these low detection rates.” Furthermore, both for patients with etiologic agent identified and no etiologic agent identified, the median period between symptom onset and testing for the samples were 2 days without difference. (Page 12 Lines 281-283)

2. The data for influenza virus should be split by at least influenza A versus influenza B (these

must surely have been differentially identified by the PCR). The nature of these two virus species is known to be different with differential impact on different ages as an example.

[Response] Many thanks for the reviewer's valuable suggestions. We have supplemented the information of genotypes of IFV (IFV-A, IFV-B, and IFV-C), RSV (RSV-A and RSV-B), and HPIV (HPIV-1, HPIV-2, HPIV-3, and HPIV-4) and comprehensively describe the genotypes of these viruses in the main text (Figs 2–3). The surveillance data and the analyses were also updated to 2019.

3) Do the authors have any suggestion as the underlying cause of the rather unusual nature of the IFV crude rate graphs; specifically the increasing rate from 0 to 18 months which is in contract to every other viral pathogen shown.

[Response] We appreciate the reviewer's comments. Previous studies using data from the national influenza surveillance program (Yu H, *et al.*, PLoS Medicine 2013) concluded similar results from the current study that school-aged children had highest positive rate of IFV (Wang Q, et al. Zhonghua Liu Xing Bing Xue Za Zhi. 2020. doi:10.3760/cma.j.cn112338-20200318-00375; Zhou L, et al., BMC Infect Dis. 2019. doi:10.1186/s12879-019-3689-9; Fang LQ, et al. Am J Epidemiol. 2012. doi:10.1093/aje/kwr411). The further Join-Point analysis conducted for IFV-A and IFV-B separately revealed an earlier age-specific peak of IFV-B than IFV-A, which was also consistent with previous studies (Wang Q, et al. Zhonghua Liu Xing Bing Xue Za Zhi. 2020. doi:10.3760/cma.j.cn112338-20200318-00375; Caini S, *et al.* PLoS One. 2019. doi:10.1371/journal.pone.0222381). A study from New York City also found that influenza epidemic period peaks occurred earliest among school-aged children each season regardless of circulating influenza viral type, subtype or strain (Olson, D. R. *et al.* PLoS Medicine, doi:10.1371/journal.pmed.0040247 (2007).). All these evidences disclosed a higher positive risk in school-age children, which may “potentially due to their lower levels of immunity than adults, fewer prior IFV infections, as well as, more opportunities for transmission to occur in highly crowded school settings compared to other community settings”. We have provided the discussion in the revised manuscript (Page 10-11, Lines 239-248).

4) In figure 4b, what does the size of the circle denote?

[Response] Thanks for the reviewer's comments. We have used color of the circles to denote the interactions between pathogens, and the circles were equal size and denote no specific meanings.

REVIEWERS' COMMENTS

Reviewer #1 (Remarks to the Author):

Although the authors have adequately addressed the reviewers' points, my major conclusion that this work belongs in a specialist journal is unchanged. However, the response was comprehensive and the methodology is now more detailed with information on genotypes from the genetic analysis, although the methods mention that sequencing was conducted but no sequence analysis is shown. The specific primer and probe sequences are also now provided. Assessing the remainder of the comments is made more difficult by failure to provide a "tracked" version but it does seem that responses to the other reviewer comments were also incorporated into the revised manuscript.

Reviewer #2 (Remarks to the Author):

The authors have successfully addressed my comments.

Reviewer #3 (Remarks to the Author):

my concerns have been addressed

Responses to the reviewers:

Reviewer #1:

1. Although the authors have adequately addressed the reviewers' points, my major conclusion that this work belongs in a specialist journal is unchanged. However, the response was comprehensive and the methodology is now more detailed with information on genotypes from the genetic analysis, although the methods mention that sequencing was conducted but no sequence analysis is shown. The specific primer and probe sequences are also now provided. Assessing the remainder of the comments is made more difficult by failure to provide a "tracked" version but it does seem that responses to the other reviewer comments were also incorporated into the revised manuscript.

[Response] Many thanks for the reviewer's positive comments on the round of revisions of our manuscript and the helpful comments, which have really improved our manuscript. In relation to "the methods mentioned that sequencing was conducted but no sequence analysis is shown", we apologize that we have not clarified this issue sufficiently. Since no sequence information was included in the current results, we have removed the description on sequencing method in this revised version of our manuscript. The subtypes of influenza viruses, RSV, and HPIV were identified by RT-PCR method with specific primers and probes. The information of all these specific primers and probes has been provided fully in the supplementary appendix (Supplementary Table 6). In addition, we apologized the inconvenience to you for not providing a "tracked" version due to more supplementary analyses and updated data in the last round of revisions according to your and the other two reviewers' suggestions and comments, only replaced by a summary of the major changes.

Reviewer #2:

1. The authors have successfully addressed my comments.

[Response] Many thanks for the comments by the reviewer.

Reviewer #3:

1. My concerns have been addressed.

[Response] Many thanks for the comments by the reviewer.